# MetaOthello: A Controlled Study of Multiple World Models in Transformers

**Aviral Chawla** [*1]  **Galen Hall** [*2]  **Juniper Lovato** [1]

## Abstract

Foundation models must handle multiple generative processes, yet mechanistic interpretability largely studies capabilities in isolation; it remains unclear how a single transformer organizes multiple, potentially conflicting "world models". Previous experiments on Othello-playing neural networks test world-model learning, but focus on a single game with a single set of rules. We introduce *MetaOthello*, a controlled suite of Othello-like games with shared syntax but different rules or tokenizations, and train small GPTs on mixed-variant data. We show that transformers trained on multiple Othello variants learn **shared world-state representations**: linear probes trained on one game intervene on another's board state nearly as well as matched probes. When the games conflict, the model resolves the resulting *ambiguity* through a localized mechanism we identify and steer. For isomorphic games with token remapping, representations are equivalent up to a single orthogonal rotation that generalizes across layers, showing the shared structure is abstract rather than tied to surface form. Together, these results show that transformers reconcile conflicting world models by sharing structure and localizing conflict. *MetaOthello* thus offers a path toward understanding how transformers organize many world models at once.

## 1. Introduction

Large transformer models trained on diverse data must learn multiple generative processes (Sikarwar et al., 2022). One model should translate languages, debug code, solve arithmetic, and it must determine which sort of text to simulate based on context (Merullo et al., 2024; Todd et al., 2024).

Mechanistic interpretability has made strides in understanding how models represent individual concepts (Park et al., 2025), and implement specific tasks (Wang et al., 2022), including evidence of emergent internal state-tracking ("world models"). Yet these analyses study one task at a time. Real models face heterogeneous data where multiple rule systems coexist—and may conflict. This raises a question: how does a single transformer organize multiple world models within a shared representation space in Othello-GPT and related systems (Li et al., 2024; Nanda et al., 2023)?

Addressing this question in large-scale models is challenging due to the difficulty of isolating specific rule systems in naturalistic data. We address this gap with *MetaOthello*, a toy-model framework for generating variants of Othello that share a common syntax (an $8 \times 8$ board) but differ in their underlying physics (update rules and validity logic). We train small transformers on "pure" datasets generated by a single game, and "mixed" datasets generated by sampling from a pair of variants.

Our main findings are:

**Cross-variant alignment.** Transformers trained on heterogeneous game data do not partition capacity into isolated sub-models. Board-state representations learned for one game transfer causally to others: linear probes trained on one variant intervene on another's internal state with effectiveness approaching matched probes.

**Syntax invariance.** For isomorphic games with scrambled tokenization, representations are equivalent up to a single orthogonal rotation that generalizes across layers—showing the model learns abstract structure independent of surface tokens.

**Economization and causal routing.** The model shares representation where games agree and diverges only where rules conflict, in proportion to the probability of conflict at each board location. A localized mid-layer circuit constructs and routes game identity: steering it causally controls which rule system the model applies to an ambiguous prefix, while a matched control does not. When variants diverge sharply, the model commits to one of them early rather than maintaining both.

These results suggest transformers reconcile multiple world models not by partitioning capacity but by sharing aligned

---
[*]Equal contribution  [1]Vermont Complex Systems Institute, University of Vermont, Burlington, VT [2]University of Michigan, Ann Arbor, MI. Correspondence to: Aviral Chawla <aviral.chawla@uvm.edu>.

*Proceedings of the $43^{rd}$ International Conference on Machine Learning*, Seoul, South Korea. PMLR 306, 2026. Copyright 2026 by the author(s).

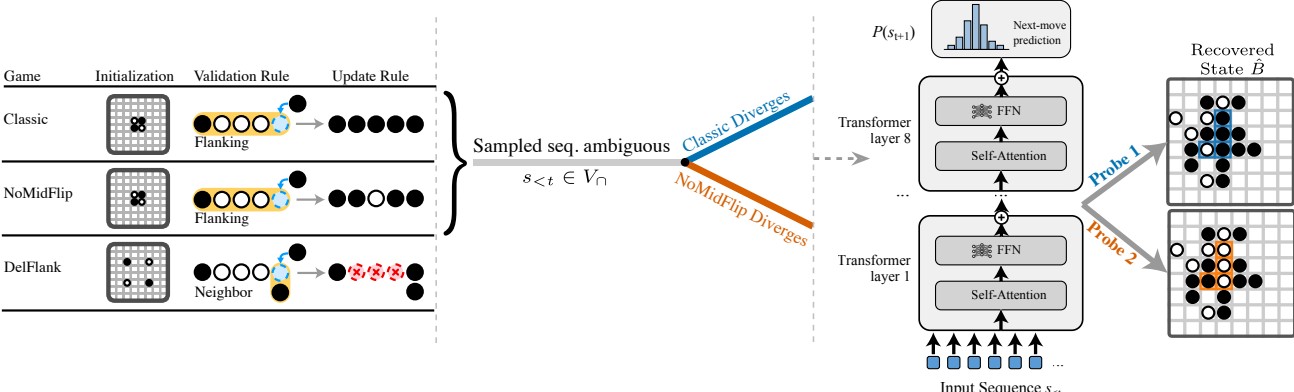

*Figure 1.* **The MetaOthello Framework.** (Left) We define a universe of games sharing a board size and vocabulary but differing in dynamics. (Middle) We sample sequences from these games. Early in the game, sequences are often valid under both rule sets (the "Ambiguous" branch), creating an informational conflict for the model. (Right) We train a small GPT model on these sequences and use Linear Probes on the residual stream to reconstruct the internal board representation.

state representations and localizing conflict to identifiable circuits. We view MetaOthello as a methodological contribution: a controlled, ground-truth testbed for developing and validating interpretability techniques that may eventually scale to how larger models organize heterogeneous knowledge.

## 2. Related Work

**Emergent world models in sequence models.** Whether sequence models learn causal world representations or merely surface statistics remains debated, but recent work has converged on a working definition that dissolves part of the disagreement: a *world model* in the interpretability sense is an internal representation sufficient to reconstruct the latent state of the data-generating process, which is distinct from the explicit forward-dynamics model of the reinforcement-learning literature (Millière & Buckner, 2024; Mitchell, 2025). We adopt this representational sense throughout.

The canonical evidence comes from Othello. Li et al. (2024) showed that probes can decode true board states from the activations of a model trained only on move sequences, and that interventions on those activations causally shift its predictions. Nanda et al. (2023) simplified this, showing linear probes suffice once tiles are coded as "mine/yours/empty" rather than absolute colors, enabling intervention by simple vector addition—the probe-and-steer methodology we build on. The finding has proven robust rather than idiosyncratic: it replicates across architectures and training setups (Yuan & Søgaard, 2025), admits circuit-level decomposition via sparse autoencoders (Wong, 2025), and supports causal interpretations of attention (Rohekar et al., 2025). Comparable state-tracking representations recur well beyond Othello—in chess-playing models (Karvonen, 2024)

and in LLMs' representations of space and time (Gurnee & Tegmark, 2023). That the same linear, causally-relevant geometry appears across games, architectures, and scales suggests it reflects a general property of how sequence models track latent state. *MetaOthello* extends this regime to a confluence of world models.

**Multi-task representation learning.** How one set of weights encodes many tasks is an emerging question. The superposition hypothesis suggests models compress features into polysemantic neurons (Elhage et al., 2022). Task vectors show distinct skills correspond to separable directions in weight space that compose arithmetically (Ilharco et al., 2023); function vectors extend this to activation space during in-context learning (Todd et al., 2024). Vafidis et al. (2025) show multi-task agents develop representations where latent factors align along orthogonal directions. The Platonic Representation Hypothesis conjectures that diverse training drives convergence toward geometrically isomorphic representations across tasks (Huh et al., 2024).

Most relevant to our work, Hua et al. (2024) train on multilingual Othello sequences (mOthello), finding cross-vocabulary alignment requires shared "anchor tokens." However, mOthello varies vocabulary while holding rules fixed—testing cross-lingual transfer, not multi-rule arbitration. MetaOthello fills this gap by varying the rules themselves: the same move sequence produces different board states under different update rules, creating genuine conflicts in latent state that neither world-model nor multi-task research has previously examined.

## 3. MetaOthello

MetaOthello is a framework defining a class of games played on an $8 \times 8$ board in which each of the 64 tiles

can have one of three states at any time: white, black, or empty. We label a unique state of the board $B \in \mathcal{B} = \{\text{white}, \text{black}, \text{empty}\}^{64}$. The set of possible moves for the current player, including 'pass' or $\varnothing$, is $M = \{\varnothing, (1, 1), \ldots, (8, 8)\}$. Since the board has 64 tiles, the move set has size $|M| = 65$: the 64 board positions plus the pass move $\varnothing$. A game, g, which is an instance of MetaOthello, is defined as the triplet $g = (B_0, V, U)$ where $B_0$ is a starting board state, $V$ is a validation rule, and $U$ is an update rule. A validation rule maps board states to sets of legal next moves: $V : \mathcal{B} \mapsto P(M)$, where $P(M)$ is the power set of $M$. An update rule maps a move applied to a board state to the resultant board state (for instance, after flanked chips are flipped): $U : M \times \mathcal{B} \mapsto B$.

We call the set of all possible sequences generated by a game $S(g)$. Given a game $g$ and a sequence of moves $s \in S(g)$, we can refer to the board state $B_k$ after these moves as $B(s, g)$. If $s \notin S(g)$ then $B(s, g) = \varnothing$. We use a shorthand to refer to the set of legal moves at turn $k + 1$ after sequence $s$ in $g$ as $V(s, g) = V(B(s, g))$, where again $V(\varnothing) = \varnothing$. (Here $V$ is applied either to a board state or, via this shorthand, to a (sequence, game) pair; the argument type disambiguates.)

### 3.1. MetaOthello Variants

We introduce two Othello variants to create mixed-game datasets: one highly similar (high game-tree overlap) and one dissimilar:

*NoMidFlip:* Same initialization and validation rules as Classic, but the update only flips the outermost two flanked tiles.

*DelFlank:* Open spread initialization instead of central; this variant employs a *neighbor validation rule*: a move is valid if and only if it is adjacent to at least one piece of the current player's color; and the update differs in that flanked pieces are deleted rather than flipped.

These variants are visualized in Figure 1. Both games generate sequences of moves that are illegal in standard Othello. NoMidFlip behaves similarly to standard Othello: the number of playable moves rises and then falls as the board fills up. DelFlank allows for games much longer than 60 because tiles can be deleted, and its game tree is consequently much larger at every depth. In either case the termination condition can trigger prior to 60 moves.

*Iago:* To distinguish between the model's ability to learn specific game rules versus its ability to form abstract structural representations, we introduce a scrambled variant of Othello. We define the Token Space $\mathcal{T} = \{0, 1, \ldots, 63\}$ as the vocabulary of the model, distinct from the Move Space $M$ (the geometric coordinates on the board). A game $g$ is now equipped with a Syntax Map $\phi_g : \mathcal{T} \to M$, a bijection that maps an arbitrary token $t$ to a physical move $m$.

We define the Iago experimental setup as a pair of games $G_{\text{Iago}} = \{g_{std}, g_{scr}\}$ which share identical logic but possess orthogonal syntax maps. This allows us to test whether the model learns a shared latent world model $W$ such that $W(s_{\text{classic}}) \approx W(s_{\text{iago}})$ despite the disjoint input vocabularies.

## 4. Model Training & Evaluation

### 4.1. Synthetic Data

We create *pure* and *mixed-game* datasets by sampling sequences either from a single game or from one of two games. In all tests, the games used are Classic, NoMid-Flip, DelFlank, and Iago. All game sequences are capped at $T_{max} = 60$. We train single-game models on datasets of $20M$ sequences and mixed-game models on datasets of $40M$ sequences ($20M$ from each).

### 4.2. Model Parameterization

We follow previous work on Othello-GPT by using an 8-layer Transformer ($L = 8$) with 8 attention heads per layer ($H = 8$) and an embedding dimension of $d_{\text{model}} = 512$ (Li et al., 2024). The context window size ($T$) was set to 59 tokens, corresponding to the maximum length of a transcript in our dataset (60).

We train the specified decoder-only Transformer model for 250 epochs each. Table 1 shows each trained model's performance. To enable fair comparison of performance across different games, we created a tailored performance metric described in the following section.

### 4.3. Evaluation

We propose a metric for model performance $\alpha(\theta \mid s)$ that (a) accounts for the best possible performance given an input sequence $s$, which would be a loss just equal to the entropy $H_G(s)$, i.e. zero excess entropy; and (b) also accounts for the loss due to random guessing, i.e. the excess entropy if the model were purely random. The second requirement arises because some games have larger next-move sets on average, and a random predictor will be inaccurately rated as higher-performing in those games unless we adjust for the size of the valid move set.

$$\alpha(\theta \mid s) = 1 - \frac{D_{\text{KL}}(P \| Q_\theta)}{D_{\text{KL}}(P \| U)} \quad (1)$$

where $U$ denotes a uniform distribution over the entire move set $M$. This normalized KL divergence is equal to 1 when the loss is minimized and 0 when it is equivalent to a uniform distribution, i.e., random guessing. The $\alpha$ score bears the additional benefit that it can compare outputs $Q_\theta$ to an arbitrary complex ground truth distribution $P$ (the mixture $P_{\text{GT}}$ defined in Appendix B), e.g., in the case where $s$ comes

*Table 1.* Model Performance: We measure model performance using the $\alpha$ metric to ensure comparability across games. All models perform well, with $\alpha = 1$ indicating perfect overlap between the model's predicted probabilities and the ground truth.

| Model Name | Game | Avg. $\alpha$ Score $\pm$ CI |
|---|---|---|
| Classic | Classic | $0.995 \pm 0.002$ |
| NoMidFlip | NoMidFlip | $0.988 \pm 0.003$ |
| DelFlank | DelFlank | $0.997 \pm 0.001$ |
| Iago | Iago | $0.994 \pm 0.002$ |
| Classic-NoMidFlip | Classic | $0.991 \pm 0.003$ |
| Classic-NoMidFlip | NoMidFlip | $0.983 \pm 0.004$ |
| Classic-DelFlank | Classic | $0.992 \pm 0.002$ |
| Classic-DelFlank | DelFlank | $0.995 \pm 0.002$ |
| Classic-Iago | Classic | $0.992 \pm 0.003$ |
| Classic-Iago | Iago | $0.991 \pm 0.003$ |
| NoMidFlip-DelFlank | NoMidFlip | $0.984 \pm 0.004$ |
| NoMidFlip-DelFlank | DelFlank | $0.996 \pm 0.002$ |

from one of 2 or more games, which may have different prior probabilities $P(g_i)$.

## 5. Results

### 5.1. Model Performance & Probe Validation

We first confirm that pure- and mixed-game models exhibit the same baseline interpretability properties as standard Othello-GPT. All models achieve near-optimal prediction, with $\alpha > 0.98$ across the board. In mixed-game models, performance dips slightly ($\approx 0.5\%$, see Appendix Figure 8). Second, generalizing the result from Nanda et al. 2023, we show that linear probes can accurately infer values of "mine," "yours," and "empty" at each board position for each game type (Appendix Figure 9). Probes trained on earlier layers and on mixed models show worse performance than those trained on later layers and single-game models.

### 5.2. Cross-variant alignment of board-state features

Given that mixed-game models linearly encode board states for all variants, we investigate their internal organization: does the model partition its residual stream into distinct subspaces for each game, or does it utilize a shared representational format? We first examine this question for rule-modified variants (NoMidFlip, DelFlank), then separately for the tokenization-scrambled variant (Iago), which requires distinct methodology.

#### 5.2.1. "PLATONIC" BOARD REPRESENTATION GEOMETRY

We first evaluate the similarity of the linear probes trained for each game variant. Each probe weight represents the direction in the activation space corresponding to all board features (e.g., "tile C3 is Mine"). We pair probe weights from Classic and variant probes trained on the mixed-model residual stream and calculate their similarities. Figure 2

reports averages and 95% confidence interval; individual tile-level values appear in Appendix Figure 12.

To assess whether probe weights encode the same features up to a rotation, we compute per-layer Procrustes alignment which finds the rotation that best superimposes one set of vectors onto another without rescaling or distortion (Schönemann, 1966): for each layer $\ell$, we find the orthogonal matrix $\mathbf{R}^\ell \in \mathbb{R}^{512 \times 512}$ that best aligns $W_{\text{Classic}}^\ell$ to $W_{\text{Variant}}^\ell$ via SVD of $\left(W_{\text{Variant}}^\ell\right)^\top W_{\text{Classic}}^\ell$. We then report cosine similarities between corresponding rows of $W_{\text{Classic}}^\ell$ and $W_{\text{Variant}}^\ell$. Since we compare learned parameters rather than model outputs, no train/test split is required.

For Classic vs. NoMidFlip, we observe high raw cosine similarity ($\approx 0.95$) in early and later layers where Procrustes alignment yields only modest further improvement ($\lesssim 0.05$). For Classic vs. DelFlank, raw similarity is moderate ($\approx 0.67$), but Procrustes alignment moves it to $> 0.90$ across the majority of layers. Both aligned similarities substantially exceed the random baseline ($\approx 0.68$ for Procrustes-aligned random matrices; Figure 2), confirming that the geometric correspondence reflects learned structure rather than high-dimensional coincidence.

#### 5.2.2. CAUSAL EFFICACY AND CROSS-PROBE INTERVENTION

Representational similarity alone does not prove that the model uses these features in the same way. To test the functional equivalence of these board state representations, we perform a cross-probe intervention experiment.

We replicate the causal intervention from previous work on Othello-GPT (Nanda et al., 2023; Li et al., 2024). We use the probe weights to intervene on the board state, $B$, to convince the model that the board state is altered, $B'$. We alter the board state by either flipping the piece from "mine" to "yours" or vice-versa or by erasing it from the board. Concretely, we accomplish this by adding the corresponding probe weight vector to the activations $h_l(s)$ at each layer:

$$h_l(s) \to h_l(s) + \gamma w_{i,c}^l \qquad (2)$$

where $\gamma$ is a scaling factor and $i, c$ index the tile and its post-intervention state. Following (Nanda et al., 2023) we apply this steering across all eight layers at once. [1] We then compute the error rate, the number of false positives and false negatives for top-$k$ predictions.

As shown in Figure 3, cross-probe interventions (hatched coral bars) are nearly as effective as "correct-probe" in-

---

[1]All-layer intervention maximizes steering strength but may over-intervene. $\gamma = 5$ for current work, but is not fully optimized. Single-layer ablations and potential compensation effects (McGrath et al., 2023) remain open for circuit-level analysis; Section 5.4.2 partially addresses layer specificity.

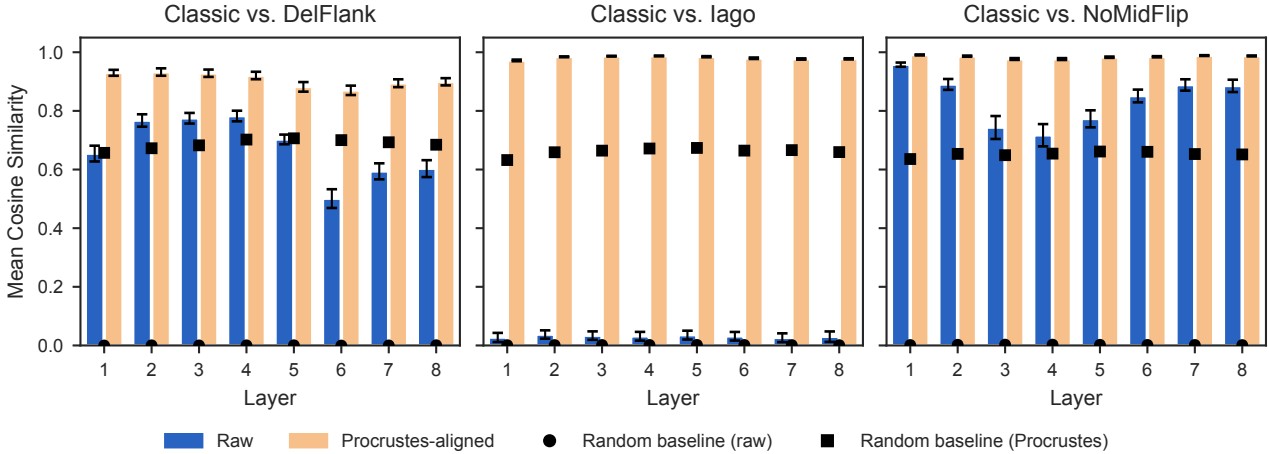

*Figure 2.* Cosine similarity between board state probe weights in mixed models, with random baseline controls. Blue bars show raw cosine similarity; orange bars show similarity after per-layer Procrustes alignment. Black circles and squares indicate expected similarity for random probes (shuffled to preserve distribution) before and after alignment, respectively. For Classic vs. Iago (center), raw similarity matches the random raw baseline (0.03). However, after alignment, similarity reaches 0.98—substantially exceeding the random Procrustes baseline of 0.68. Error bars denote 95% CIs across 192 probe dimensions.

terventions (blue bars) at steering the model's predicted board state. This dissociation between raw representational similarity and causal efficacy is notable: even when probe similarity is moderate (as in Classic–DelFlank), the Classic probe can intervene on DelFlank boards with comparable accuracy.

### 5.3. Syntax Invariance: The Iago Experiment

The preceding analyses examined games sharing tokenization but differing in rules. We now ask the complementary question: when games are computationally isomorphic but use permuted token vocabularies, does the model still learn a shared latent representation?

At a first glance, raw cosine similarity between Classic and Iago probes is indistinguishable from a random baseline ($\approx$ 0.03; Figure 2, center). However, after per-layer Procrustes alignment, similarity reaches 0.98—substantially exceeding the random-alignment baseline of 0.68. The model learns the same latent structure for both games, differing only by a rotation induced by the input vocabularies.

We now test whether this rotational equivalence on probe weights extends to the full residual stream. We estimate a single global transformation $\Omega \in \mathbb{R}^{512 \times 512}$ via orthogonal Procrustes on paired Classic/Iago activations, pooling across all layers and positions, on a training set.

We then feed a test set of Classic sequences into the mixed model, apply $\Omega$ at a specific intervention layer $\ell'$, and measure performance on corresponding Iago moves. As shown in Figure 4, intervening at any layer (except for 8) recovers near-perfect performance on Iago moves; far above the

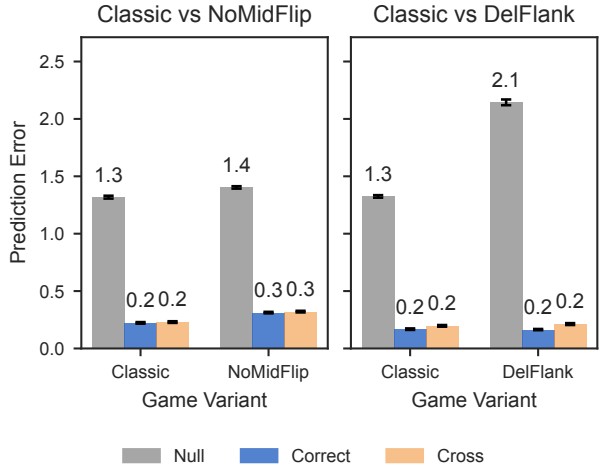

*Figure 3.* Global intervention error across conditions. Gray bars show null baseline (no intervention); blue bars show correct-probe intervention; orange bars show cross-probe intervention. Error bars denote 95% CI. We see that cross-probe intervention is nearly as effective as the correct probe in steering board states.

$\alpha = -2.9$ obtained without the rotation.

These results suggest that the transformer's board-state representations are largely syntax-invariant up to a token permutation.

### 5.4. How do the models handle ambiguous sequences?

Although Classic and DelFlank/NoMidFlip board representations differ considerably for particular tiles, states, and layers, intervening on a variant's representation still influences

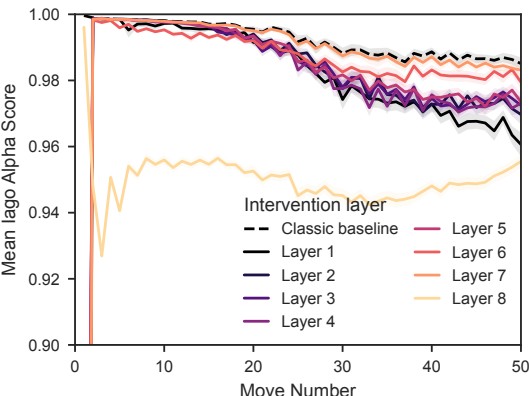

*Figure 4.* Classic-to-Iago activation alignment via orthogonal Procrustes. We feed Classic sequences to the mixed Classic-Iago model, apply a learned orthogonal rotation $\Omega$ to the residual stream at layer $l'$, and measure how well the model predicts corresponding Iago moves ($\alpha$ score). See Appendix C.1 for layer 8 inconsistency.

Classic next-token predictions about as well as intervening on the Classic representation. In the mixed-game context—particularly Classic-NoMidFlip—this raises a conundrum. Many sequences $s^*$ are present in the languages of both games yet generate different board states and valid move sets:

$$B(s^*, g_1) \neq B(s^*, g_2), \qquad V(s^*, g_1) \neq V(s^*, g_2).$$

We call these *ambiguous* sequences. To perform well on them, the model must predict a weighted combination of next tokens drawing on *both* games' board states, even though prior results suggest a single *causal* board state representation. Optimal next-token prediction under the cross-entropy loss is:

$$P(x \mid s^*, \{g_1, g_2\}) = P(g_1 \mid s^*)\frac{\mathbb{1}[x \in V(s^*, g_1)]}{|V(s^*, g_1)|}$$
$$+ P(g_2 \mid s^*)\frac{\mathbb{1}[x \in V(s^*, g_2)]}{|V(s^*, g_2)|}. \quad (3)$$

How, then, does the model represent the board state(s) and manage potentially conflicting representations?

### 5.4.1. STEERING VECTORS GOVERN LAYERWISE SPECIALIZATION

The Classic–NoMidFlip games share validation rules and board dimensions but differ in update dynamics, creating widespread ambiguity: sequences remain valid in both games until a move triggers the divergent update. We quantify this as the average entropy of game identity given a sequence, $\mathbb{E}[H(g \mid s_{<t})]$ (Figure 5b, inset).

While Section 5.2 showed Classic and NoMidFlip representations are causally interchangeable, they are not identical.

For each tile, we compute the probability its value differs between the two games on a randomly sampled ambiguous $s^*$, and find that representational dissimilarity correlates strongly with this conflict probability—increasingly so at higher layers, reaching $R^2 = 0.95$ at layer 8, with a sharp rise after layer 5 ($R^2_{L5} = 0.35 \rightarrow R^2_{L6} = 0.73$; Appendix Figure 10). Comparing principal angles between each position's $(v_{mine}, v_{yours})$ subspaces gives qualitatively similar results across all layers ($R^2 \in [0.70, 0.89]$). As expected, "Empty" representations are nearly identical across games, since playing a tile permanently sets a non-empty value. Thus the model economizes on dual world models, allocating extra capacity where the two worlds are likely to diverge. Consistent with this, NoMidFlip board probes recover conflicting tile values with accuracy that declines monotonically with move number, and the layer-5 probe is most accurate throughout (Figure 5a).

Continuing to ground causal analysis in linearly decodable representations, we ask whether the model linearly encodes the likelihood of each game given a sequence. We train a probe to predict the Bayesian ground-truth likelihood $P(g \mid \tilde{s})$ from $h_l(\tilde{s})$ at each layer, comparing against a baseline probe on $60 \times 60$ one-hot move-and-move-number vectors that captures the game-identity information present prior to any model processing. Layers 1–4 perform no better than baseline, while layers 5–8 carry a reliably 90% accurate representation of game likelihood (Figure 5b). When a move exits the Classic ∩ NoMidFlip tree into Classic \ NoMidFlip, the inferred probability of NoMidFlip drops to zero most reliably in layers 5–6, though this responsiveness decays with sequence length (Figure 11).

We test the causal role of these game probabilities by adding $\Delta\mu = \mu_{\text{NoMid}} - \mu_{\text{Classic}}$, derived from the game-ID probe weights, to the residual stream of sequences valid under both games (Figure 5c). Injection at early (1–4) and late (6–8) layers has little effect, but intervention at Layer 5 significantly increases NoMidFlip-consistent moves—while only negligibly changing downstream NoMidFlip board-state fidelity, even on the conflicting tiles. Layer 5 thus acts as a pivot where shared representations diverge into game-specific computation.

### 5.4.2. MECHANISMS OF DISAMBIGUATION

The $\Delta\mu$ intervention localizes the decision to Layer 5 but treats it as a single direction; we now probe for the exact mechanisms. We track the signed game-ID margin (probe logit for the correct game minus that for the incorrect one) decomposed across activations 1) before attention, 2) after attention, and 3) after the MLP of each layer (Figure 6a). Layers 1–3 carry only a weak game-ID margin ($\approx 19\%$ of the post-layer-4 total). The first substantial gain comes from the layer-4 MLP, which amplifies this upstream signal

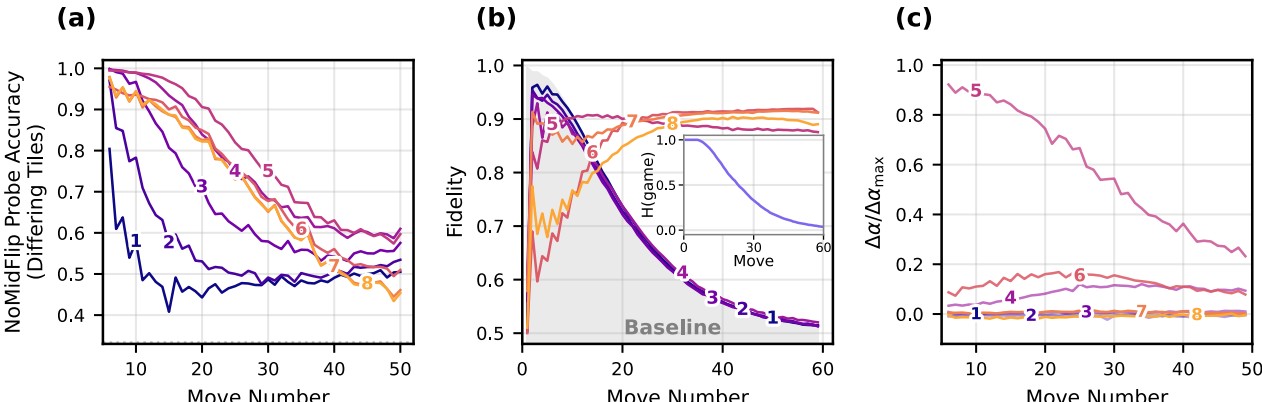

*Figure 5.* **Differentiation dynamics in the Classic–NoMidFlip mixed model.** (a) performance of NoMidFlip probes on tiles that differ between NoMidFlip and Classic after an ambiguous sequence $s^*$. (b) Probe Fidelity: Probes trained to estimate the probability of the game context ($P(\text{Classic})$) where fidelity is $(1 - |P_{\text{probe}} - P_{\text{GT}}|)$. We also train a "Baseline" probe on one-hot encoded $60 \times 60$ (move $\times$ move number) inputs. Inset: Average entropy of the ground truth game distribution $P(g|s_{<t})$ over time. (c) Causal Steering: Injecting a game-steering vector ($\lambda[\mu_{\text{NoMid}} - \mu_{\text{Classic}}]$) to measure the change in adherence to NoMidFlip rules, measured by the normalized increase in $\alpha$-score relative to NoMidFlip-valid moves.

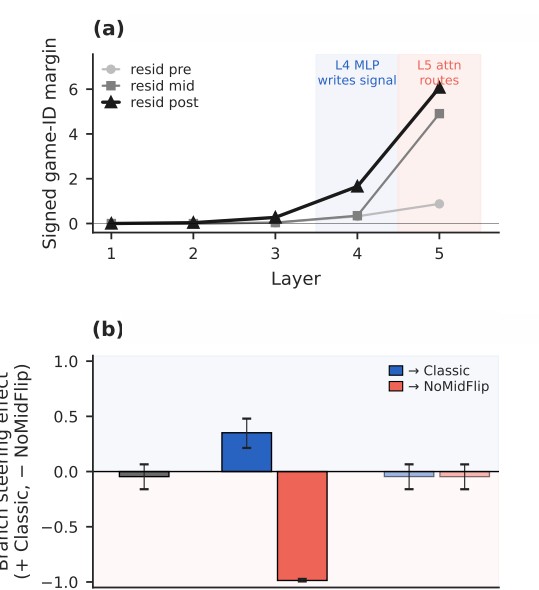

*Figure 6.* **Causal routing resolves ambiguous Classic–NoMidFlip prefixes.** (a) A stage-wise readout of the signed game-ID margin at `resid_pre`/ `mid`/ `post` shows that early layers carry only a weak signal; the first substantial gain follows the layer-4 MLP and the largest jump follows layer-5 attention, indicating game identity is predominantly built at the layer-4/layer-5 transition. (b) Steering the identified circuit on held-out ambiguous prefixes, before any disambiguating move. We measure Classic's share of probability mass on moves valid in exactly one game (0.5 = no branch preference). Full-circuit steering shifts the model into the selected branch in both directions, while the runner-up layer-5 head control stays at baseline. The circuit is thus not merely diagnostic of game identity; it causally routes ambiguous prefixes into game-specific computation.

roughly fivefold; the layer-4 MLP is thus the dominant contributor to the game-ID feature, sharpening a faint early signal rather than building one from a fully game-agnostic state. The largest single jump is then contributed by layer-5 attention. This points to a two-part picture: the layer-4 MLP writes most of the disambiguating signal, and layer-5 attention routes it into a direction available downstream.

We localize the responsible components by zeroing each component's last-token output and measuring the resulting drop in the layer-5 game margin. The layer-4 MLP produces the largest effect ($\Delta = -3.97$; this includes the downstream cascade, since starving the layer-4 MLP also deprives layer-5 attention of a feature to route), followed by layer-5 attention ($-1.58$) and the layer-5 MLP ($-1.16$). Within layer-5 attention, a single head dominates: head 5 yields $\Delta = -1.77$ against $-0.17$ for the next-strongest head, and across 1,000 bootstrap resamples over ambiguous prefixes it is the top head in $100\%$ of samples. The disambiguation circuit is therefore a compact three-stage path—layer-4 MLP $\rightarrow$ layer-5 head 5 $\rightarrow$ layer-5 MLP—and we use the runner-up layer-5 head as a strong control in the steering tests below.

We next characterize what the layer-4 MLP computes. For each tile we form the direction, within the correct game's board probe, separating the true tile state from the alternate game's tile state, and project the layer-4 MLP output onto it. This projection is sharply selective: it is large on tiles where Classic and NoMidFlip imply different states and near zero elsewhere ($2.16$ vs. $\approx 0$; rank-biserial $0.82$, Mann–Whitney $p < 10^{-6}$), and it is symmetric across the two branches ($2.19$ on Classic-only prefixes vs. $2.14$ on NoMidFlip-only). Moreover, the per-tile projection magnitude tracks the em-

pirical probability that a tile's value disagrees between the games (Pearson $r = 0.975$)—the layer-4-MLP analog of the residual-stream result of Section 5.4.1, where representational dissimilarity tracked the same disagreement probability at $R^2 = 0.95$. The MLP thus computes a *rule-conflict feature* from the shared board representation, writing game-distinguishing information precisely on the tiles where the two worlds diverge.

Finally, we test whether this circuit is causally sufficient to set the branch. We learn each component's Classic-vs-NoMidFlip direction on one set of ambiguous prefixes and steer a disjoint held-out set, before any disambiguating move (Figure 6b). Steering the full circuit moves Classic's share of unique-move mass from $0.48$ at baseline to $0.68$ in the Classic direction and $0.01$ in the NoMidFlip direction, scaling monotonically with steering strength; steering the routing head alone is weaker, and the runner-up head with its own learned direction leaves the share at baseline. The effect is also behavioral but concentrated in rare cases: ablating the circuit barely changes headline valid-move mass ($0.999 \rightarrow 0.964$), yet increases mass on moves valid *only* in the wrong game by roughly two orders of magnitude ($0.0001 \rightarrow 0.0115$), exactly the minority of inputs that depend on correct routing.

The above results explain how shared and game-specific representations coexist. We see cross-probe interventions as interchangeable (Figure 3) because the board representation is genuinely shared. Then, a thin, separately-computed conflict feature makes the two games' predictions diverge. The model does not choose between one shared world model and two isolated ones; it economizes, carrying common state in the shared substrate and resolving conflict through a localized routing circuit that determines which rule system applies where the games disagree.

### 5.5. Out-of-distribution world modeling

The Classic–NoMidFlip mechanism operates where ambiguity is common: the two games stay jointly valid for many moves, so the model is rewarded for maintaining both board states and routing between them. Classic–DelFlank is the opposite regime. DelFlank's permissive neighbor-validation rule and its ability to delete tiles make its game tree scale exponentially with move number, whereas the Classic and NoMidFlip trees scale subexponentially outside the mid game. Consequently, although Classic $\cap$ DelFlank is nonempty until at least move 50, it is a vanishing fraction of DelFlank's tree, and the posterior $P(\text{DelFlank} \mid s_{<t})$ for $s_{<t} \in \text{Classic} \cap \text{DelFlank}$ collapses rapidly: of 20M random DelFlank training sequences, we expect $\sim 19{,}500$ to remain ambiguous at move 5, $\sim 19$ at move 10, and $\lesssim 1$ beyond move 12. Ambiguous sequences past $t^* \approx 12$ are thus functionally *out of distribution*, and optimal $\alpha$ on them

requires predicting as if the sequence were Classic, even though it remains DelFlank-legal.

We first ask whether the model nonetheless maintains a distinct DelFlank board representation on such sequences. Such a representation would be useful only in vanishingly rare cases. Assessing the DelFlank board probes (high-fidelity on random DelFlank games) on these intersection sequences, we find Layer-5 accuracy drops to $\approx 67\%$, versus $\approx 99\%$ for Classic probes on the same inputs. Rather than encoding both boards as in the NoMidFlip case, the model has effectively *selected* the Classic world model and let the DelFlank representation degrade.

We then test whether this selection can be reversed by intervention, applying a game-identity steering vector $\Delta\mu = \mu_{\text{DelFlank}} - \mu_{\text{Classic}}$ analogous to the NoMidFlip case (Figure 7). The effective intervention site moves: steering at early layers (2–3) drives a strong shift toward DelFlank-valid moves ($\Delta\alpha \approx 0.75$ at low move numbers, decaying to $\approx 0.3$ by move 30; Panel A), while Layer 5 has essentially no effect ($\Delta\alpha \approx 0$). Early-layer steering also restores the *downstream* representation (Panel B): the Layer-5 DelFlank probe rises from $67.1\%$ to $77.4\%$ after Layer-2 steering ($+10.3\%$) and $75.8\%$ after Layer-3 steering ($+8.7\%$).

These two effects distinguish DelFlank from NoMidFlip. We find an early rather than mid-layer intervention site, and a causal improvement in downstream board fidelity. The contrast suggests two qualitatively different arbitration regimes. When competing rule systems remain jointly plausible (NoMidFlip), the model maintains both boards and *reweights* a shared mid-layer subspace at the point of conflict. When one system becomes effectively out of distribution (DelFlank), the model commits early, *selecting* a single world model and discarding the alternative, such that recovering the suppressed interpretation requires intervening early.

## 6. Conclusion

This study leverages the MetaOthello framework to show that transformers trained on heterogeneous generative processes do not partition their capacity into isolated sub-models. Instead, they converge on a shared representation of state that is efficiently reused across contexts: board-state representations are geometrically aligned across variants, with linear probes transferring causally between games with different update rules, and for syntax-scrambled variants (Iago) they are identical up to a single orthogonal rotation that generalizes across layers.

This shared substrate raises an apparent paradox: if the two games' representations are causally interchangeable, how does the model still distinguish them on ambiguous prefixes where the rules eventually diverge? We resolve this

**(a)**

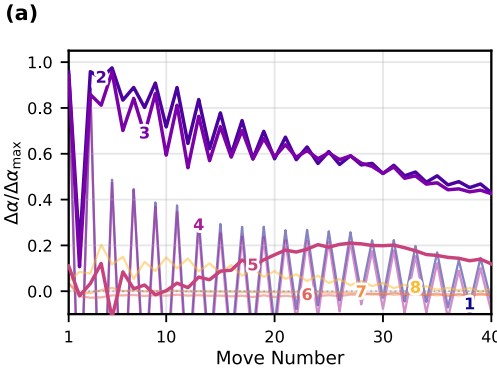

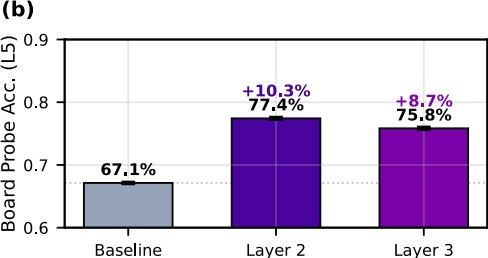

*Figure 7.* **Steering DelFlank on ambiguous sequences.** (A) Normalized improvement in $\alpha$-score toward DelFlank-valid moves ($\Delta\alpha/\Delta\alpha_{\max}$) as a function of move number. Early layers (2–3) show strong steering effects that decline with game length; Layer 5 shows no effect. (B) DelFlank board probe accuracy at Layer 5 under different steering conditions. Steering at early layers causally improves downstream representations, providing evidence for a selection mechanism distinct from NoMidFlip's subspace reweighting.

by localizing the conflict-handling mechanism. The board representation is genuinely shared, while a thin, separately-computed conflict feature determines which rule system the model applies. Steering this circuit causally selects the branch on held-out ambiguous prefixes, while a matched control does not. The model thus economizes: it carries common state in the shared substrate and spends extra capacity only on localized routing where the worlds conflict.

We further find that the form of this arbitration depends on how far the rule systems diverge. When competing rules remain jointly plausible, the model maintains both board states and reweights a shared mid-layer subspace at the point of conflict; when one rule system is effectively out of distribution, it instead commits early to a single world model, so that recovering the suppressed interpretation requires intervening in early rather than middle layers. How models arbitrate between world models is therefore not a single mechanism but a regime that shifts with the distance between the competing processes.

While this work uses a toy model, it offers methodological contributions to mechanistic interpretability. By providing a controlled setting where ground-truth latent states *and*

the ground-truth conflict structure between them are known, MetaOthello lets researchers validate probing, causal intervention, and circuit-discovery techniques against a known answer before deploying them where it is hidden. Our results also caution that a model's interchangeable behavior on the inputs where world models agree reveals little about the localized machinery that governs the rare inputs where they conflict. The finding that arbitration localizes to identifiable, steerable circuits rather than diffuse processing suggests how larger models may organize heterogeneous knowledge, and motivates more robust methods for analyzing internal state-tracking in general-purpose systems.

## 7. Limitations

Our findings are established in a controlled, synthetic setting and may not fully capture real-world multi-task learning. MetaOthello systematically varies rules and tokenizations, but remains far simpler than heterogeneous corpora where rule systems are implicit and numerous. We only test 50/50 mixtures, one 8-layer ($d_{\text{model}} = 512$) architecture, and pairwise game mixtures, so the effects of imbalanced mixtures, scale, and many-way arbitration remain open. Finally, our analysis relies on linear probes; nonlinear structure and circuit-level analysis may reveal additional mechanisms.

## Acknowledgments

The authors would like to thank the reviewers for their feedback on this project. The authors would like to thank Melanie Mitchell for her feedback and comments on our analysis. Our team acknowledges support from Alfred P. Sloan Foundation (Grant #G-2024-22498), and the National Science Foundation (Grant #2242829)

## Impact Statement

This paper presents work whose goal is to advance the field of Machine Learning, specifically in the area of mechanistic interpretability. By establishing a controlled setting to study how models organize conflicting knowledge, this research contributes to the broader goal of making "black box" systems more transparent. Understanding how foundation models separate or merge different task contexts is a step toward verifying their safety and reliability in complex, real-world deployments. Beyond these contributions to model transparency and safety research, there are many potential societal consequences of our work, none of which we feel must be specifically highlighted here.

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

# A. Reproducibility

## A.1. Data and Code Availability

All code and pretrained assets are publicly available:

- **Code:** https://github.com/aviralchawla/metaothello

- **Models & probes:** https://huggingface.co/aviralchawla/metaothello

- **Training data:** https://huggingface.co/datasets/aviralchawla/metaothello

The repository includes the MetaOthello game engine, data generation and model training scripts, board probe training and intervention analysis code, and all pretrained checkpoints. See the repository README for installation and reproduction instructions.

## A.2. Training Details

**Architecture.** All models use an 8-layer decoder-only Transformer with 8 attention heads per layer, embedding dimension $d_{\text{model}} = 512$ We use learned positional embeddings with a context window of $T = 59$ tokens. The vocabulary size is 66 (64 board positions plus a skip token and a pad token).

**Optimizer.** We use AdamW (Loshchilov & Hutter, 2019) with $\beta_1 = 0.9$, $\beta_2 = 0.99$, and weight decay $\lambda = 0.01$.

**Learning rate.** We use a linear warmup over the first 1,000 steps to $5 \times 10^{-5}$ over the remaining training.

**Batch size.** We use a batch size of 4096 sequences.

**Training duration.** Single-game models are trained for 250 epochs on 20M sequences. Mixed-game models are trained for 250 epochs on 40M sequences. Training converges well before 250 epochs; we use this fixed budget for consistency.

## A.3. Probe Training Details

**Architecture.** Linear probes are single-layer linear classifiers mapping from $\mathbb{R}^{512}$ to $\mathbb{R}^3$ (for mine/yours/empty classification) per board position, yielding 64 independent 3-way classifiers.

**Training data.** Probes are trained on 100,000 sequences (sampled independently from model training data) with an 80/20 train/validation split.

**Optimizer.** Adam with learning rate $3 \times 10^{-5}$, trained for 10 epochs with early stopping based on validation accuracy (patience = 5 epochs).

**Game-identity probes.** These use the same architecture but predict $P(\text{game} \mid s_{<t})$ as a binary classification task, trained with cross-entropy loss against the Bayesian ground truth computed from Equation 5.

## A.4. Random Seeds

All models and probes use a fixed random seed (42) for reproducibility. Due to computational constraints, all models are trained only once.

# B. Model performance and $\alpha$ score

Standard metrics such as cross-entropy loss or top-1 accuracy are insufficient for evaluating model performance in the MetaOthello setting. Because the size of the valid move set $|V(s, g)|$ varies significantly depending on the game variant $g$ and the game state $s$, the intrinsic entropy of the ground truth distribution fluctuates. A loss of 2.0 nats might represent perfect performance in a high-entropy state (many valid moves) but poor performance in a low-entropy state (forced move).

To address this, we introduce the $\alpha$-score, a normalized metric that measures how much of the reducible uncertainty the model has resolved, relative to a random baseline.

## B.1. Ground Truth Distribution in Mixed Environments

In a mixed-game setting, the ground truth distribution over the next token $x_t$, given a history $s_{<t}$, is a mixture distribution marginalized over the set of possible games $\mathcal{G} = \{g_1, \ldots, g_K\}$.

First, we define the likelihood of observing a sequence $s_{<t}$ under a specific game $g$. Assuming the data generation process samples move uniformly from the set of valid moves $V(s_{<k}, g)$ at each step $k$, the likelihood is the product of the inverse valid move set sizes:

$$P(s_{<t} \mid g) = \prod_{k=0}^{t-1} \frac{1}{|V(s_{<k}, g)|} \cdot \mathbb{I}[s_k \in V(s_{<k}, g)] \tag{4}$$

where $\mathbb{I}[\cdot]$ is the indicator function, which is 0 if a move is illegal in game $g$.

We define the posterior probability that the current sequence is being generated by game $g$ using Bayes' theorem. Let $P(g)$ be the prior probability of sampling game $g$ (in our mixed datasets, $P(g) = 0.5$). The posterior is:

$$P(g \mid s_{<t}) = \frac{P(s_{<t} \mid g)P(g)}{\sum_{g' \in \mathcal{G}} P(s_{<t} \mid g')P(g')} \tag{5}$$

Substituting the likelihood from Eq. 4 we obtain the formulation described in the main text: the probability of a game is proportional to the product of the inverse valid move set sizes along the sequence.

Finally, the ground truth probability of the next token $x_t$ is the expectation over the game posteriors:

$$P_{\text{GT}}(x_t \mid s_{<t}) = \sum_{g \in \mathcal{G}} P(x_t \mid s_{<t}, g)P(g \mid s_{<t}) \tag{6}$$

where $P(x_t \mid s_{<t}, g) = |V(s_{<t}, g)|^{-1}$ if $x_t$ is valid in $g$, and 0 otherwise.

## B.2. The $\alpha$-Score Definition

We seek a metric $\alpha(\theta \mid s)$ that quantifies the model $Q_\theta$'s proximity to $P_{\text{GT}}$. We define two reference divergences:

1. **Model Divergence:** $D_{\text{KL}}(P_{\text{GT}} \| Q_\theta)$, the excess entropy of the model relative to the ground truth.

2. **Random Baseline Divergence:** $D_{\text{KL}}(P_{\text{GT}} \| U)$, the excess entropy of a uniform distribution $U$ (random guessing over the vocabulary) relative to the ground truth.

The $\alpha$-score is defined as:

$$\alpha(\theta \mid s) = 1 - \frac{D_{\text{KL}}(P_{\text{GT}} \| Q_\theta)}{D_{\text{KL}}(P_{\text{GT}} \| U)} \tag{7}$$

This metric has the following desirable properties:

- $\alpha = 1$: The model perfectly matches the ground truth mixture distribution ($Q_\theta = P_{\text{GT}}$). This implies the model has perfectly disentangled the ambiguous games or perfectly modeled the superposition.

- $\alpha = 0$: The model performs equivalent to random guessing over the vocabulary.

- $\alpha < 0$: The model is being confidently wrong (assigning low probability to valid moves).

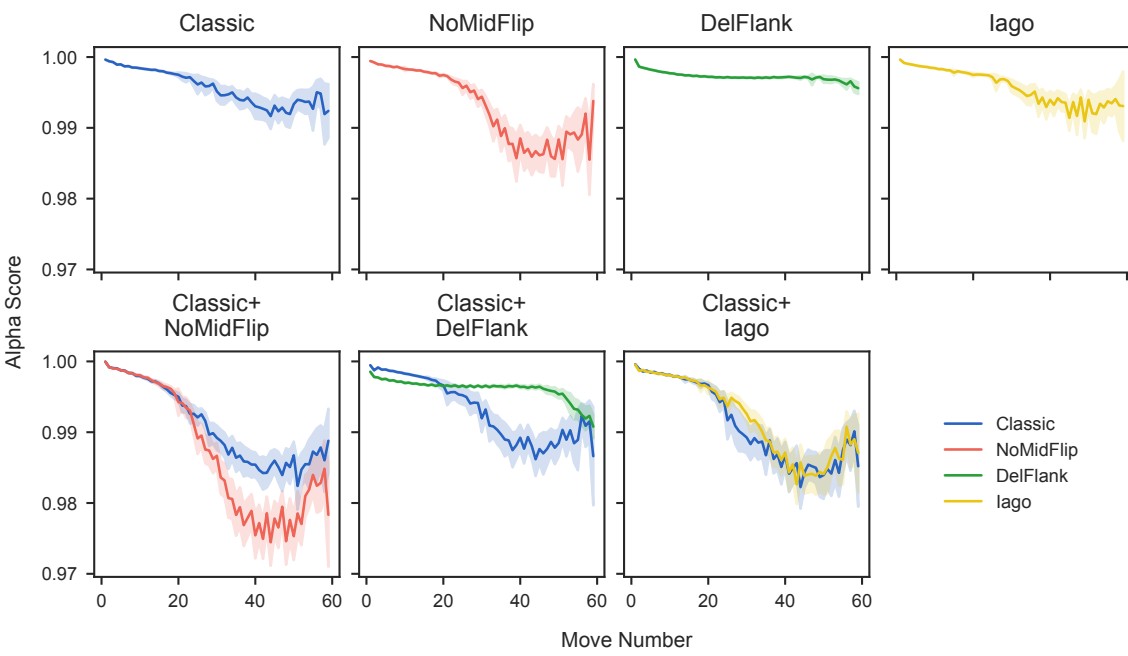

*Figure 8.* Alpha scores of all models for sequences sampled from each game, shown as a function of move number with 95% intervals.

- It is naturally extensible to other comparisons besides the uniform distribution $U$; for instance, in a mixed game setting, we could replace $U$ with the distribution of next moves according to one of the two games.

- Crucially, $\alpha$ allows us to compare performance across games with vastly different branching factors (e.g., DelFlank vs. NoMidFlip) on a unified scale.

## C. Probe performance

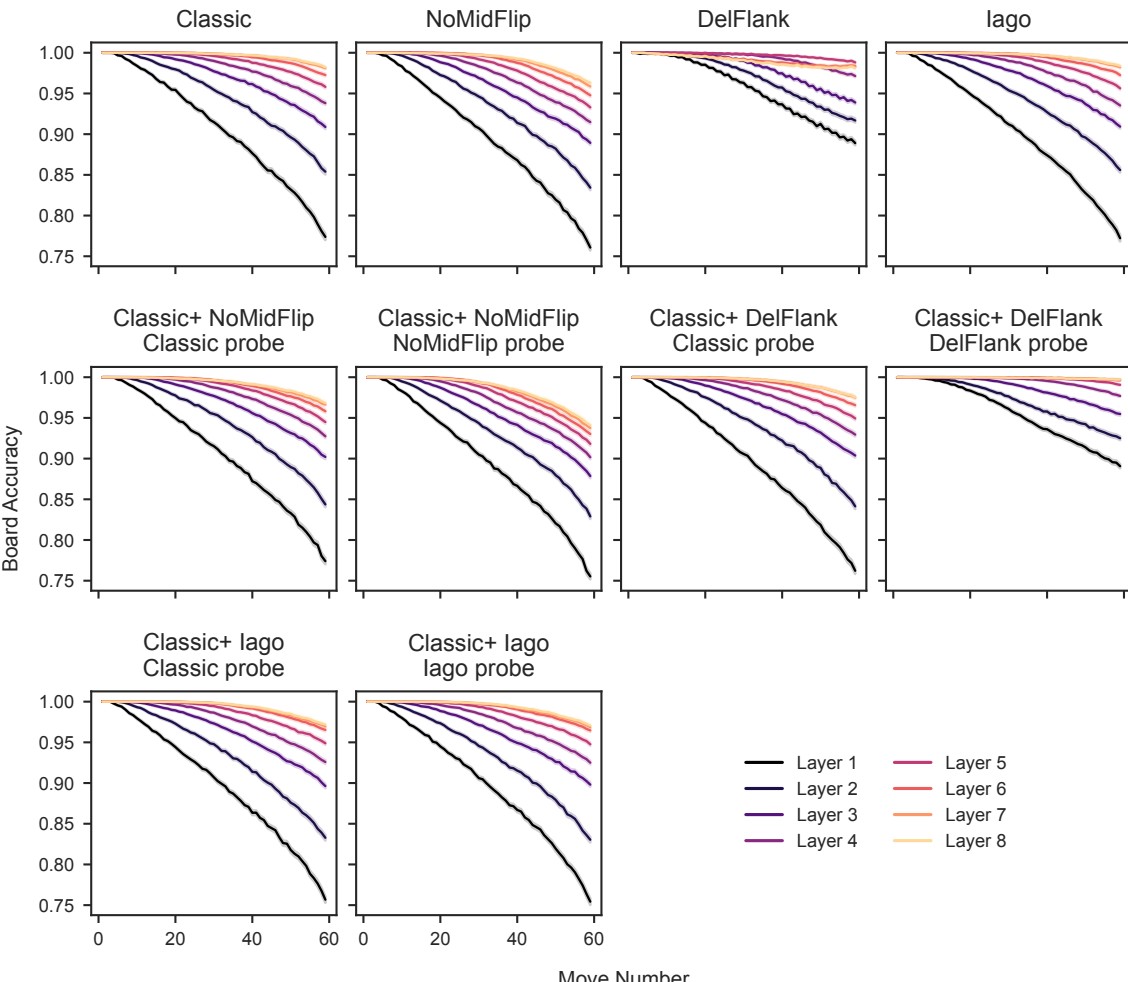

*Figure 9.* Accuracies of linear probes trained to detect board states from model activations.

### C.1. Why layer-8 intervention behaves differently across move number

Figure 4 shows that intervening with the global rotation $\Omega$ at layers 1–7 yields near-baseline Iago $\alpha$ across all move numbers, whereas the layer-8 intervention is high at the first prediction and decays with move number—the opposite trend. This follows from where the intervention sits relative to the rest of the computation, not from a qualitative change in the representation.

For layers $\ell' \leq 7$, applying $\Omega$ rotates the residual stream into Iago-aligned coordinates that the remaining $8 - \ell'$ layers then process normally; the intervention converts the ongoing computation rather than only its output, so performance is robust as the prefix lengthens. Layer 8 is the final residual stream, immediately before the unembedding: an intervention there is effectively a late output remapping and cannot redirect the upstream, Classic-syntax computation that already produced the hidden state. At the first plotted position the prediction is close to a one-step token translation, so an output remap suffices and $\alpha$ is high; as the prefix grows, the hidden state encodes more game-state structure computed in the unrotated frame, which a single terminal rotation cannot retroactively correct, and $\alpha$ falls. This is consistent with later-layer representations being more layer-specialized, so the globally-pooled $\Omega$ fits them less well than it fits early/middle layers.

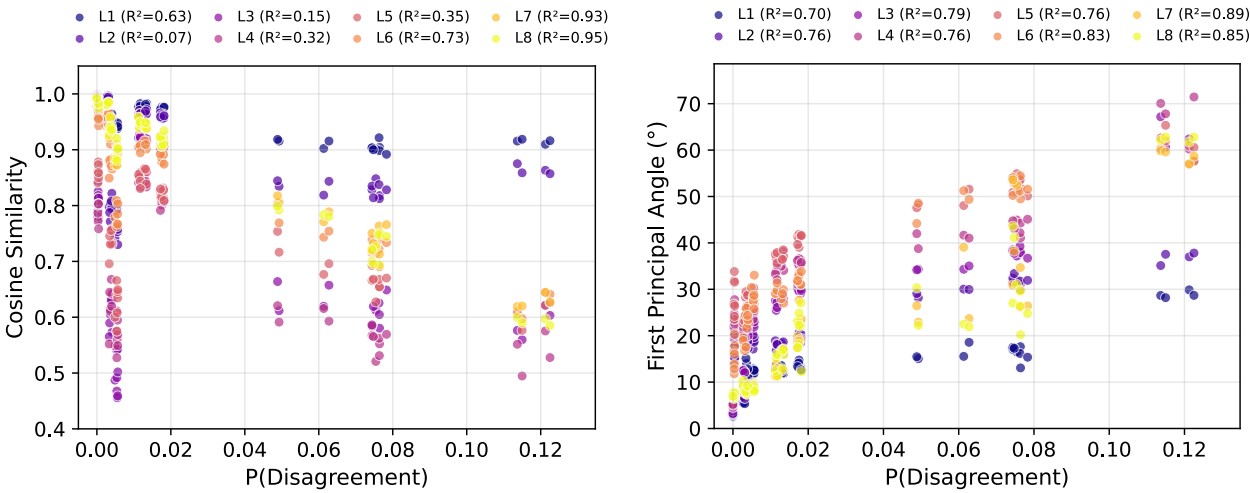

*Figure 10.* (Left) Cosine similarity and (Right) Principal angle (deg.) between Classic and NoMidFlip tile-state representations, plotted against the actual probability that a given tile will have a conflicting state under the two rules.

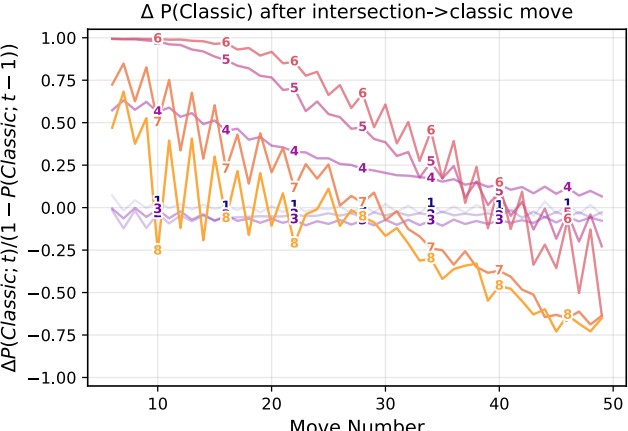

*Figure 11.* Change in inferred $P(\text{Classic}|s)$ after playing a move only legal under Classic rules at the end of an ambiguous sequence $s^*$. Under optimal behavior, the probability of NoMidFlip should collapse to 0 and the probability of Classic to 1; we ask how close the inferred probability gets, compared to its value prior to this move.

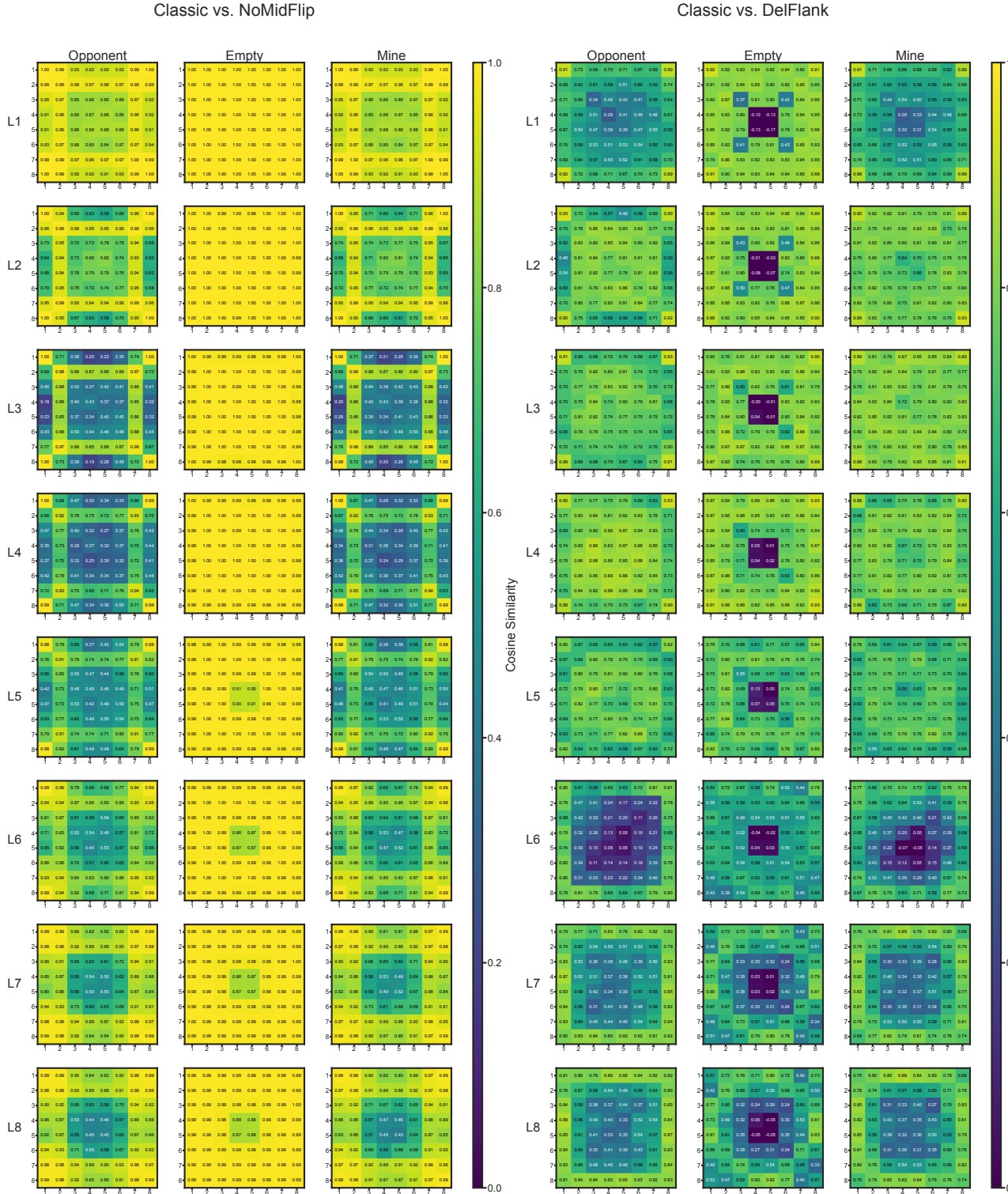

*Figure 12.* Cosine similarity of tile-state representations from each layer's probe for Classic and NoMidFlip (Left) and Classic and DelFlank (Right).

