# OpenReview forum: "MetaOthello: A Controlled Study of Multiple World Models in Transformers"
_ICML.cc/2026/Conference — ICML 2026 regular_

### Official Review · Reviewer_o2yh · 2026-03-12

**Soundness:** 4
**Presentation:** 4
**Significance:** 3
**Originality:** 3
**Overall Recommendation:** 5
**Confidence:** 3

**Summary:**

MetaOthello is a suite of 3 versions of the game Othello with the same syntax but different rules and one version with different tokenizations. These variants are then utilized to train models, with the primary goal of enhancing our understanding of mechanistic interpretability. The study explores whether transformers can adapt to multiple tasks across these different game variants. Specifically, it investigates whether this adaptation occurs through the creation of specialized sub-models for each game type or if there is another underlying representation mechanism at play. The paper states that the transformer does not rely on separate sub-models for each game variant and instead, it suggests that the early layers of the transformer maintain a game-agnostic representation, the middle layers identify the specific game type, and the later layers specialize for that particular variant. Additionally, the research demonstrates that linear probes trained on one variant can effectively transfer their knowledge to another variant, achieving performance levels comparable to those of matched probes. This framework and its findings provide valuable insights into how transformers can generalize across diverse tasks and environments.

**Compliance With Llm Reviewing Policy:**

Affirmed.

**Final Justification:**

The authors have addressed the concern I raised in the initial review. This issue was limited to presentation-level aspects and do not affect my evaluation. I therefore keep my positive score.

**Key Questions For Authors:**

I do not have specific questions for the authors. Clarification of the numbering inconsistency of layers(Fig. 6) mentioned earlier would be helpful.

**Limitations:**

Yes

**Strengths And Weaknesses:**

Strengths:

	Soundness: The study rigorously tests its hypotheses using a controlled suite of Othello variants, providing clear evidence of how transformers handle multiple generative processes. Empirical validation through linear probes and causal interventions supports the claims made in the paper.

	Presentation: The paper is generally well organized and easy to follow.

	Significance: This paper addresses the problem of mechanistic interpretability and proposes a representation that, to my knowledge, has not been explored in this form before. Although the scope is specialized for Othello variants, or rather cases where the syntax are the same with differing rules, it provides a significant contribution which can influence future research.

	Originality: The paper introduces variations on Othello games with the same syntax but differing rules or tokenization providing insights on the workings of world models in transformers. It also demonstrates original findings regarding the convergence of board-state representations and the layer wise specialization in handling conflicting rules.

Weaknesses:

	Soundness: I did not identify major weaknesses.

	Presentation: There appears to be a numbering inconsistency of the layer number mentioned in Figure 6 in the citation and at the last paragraph of section 5.6. Need clarity if this is a typo.

	Significance: I did not find any major weaknesses.

	Originality: I did not identify clear weaknesses regarding originality, although my familiarity with some related work in this area is limited.

---

> ### Author Rebuttal · Authors · 2026-03-31
>
> We thank the reviewer for a positive assessment and careful reading. The reviewer correctly identifies a layer numbering error. The last paragraph of Section 5.6 refers to layers 1 and 2 when it should say layers 2 and 3. The figure itself is correctly labeled, but we reduced the opacity of the low-signal layers’ lines, which left line 1 hard to see, while the label itself has 100% opacity. This is a cosmetic error, not a mislabeling, and we thank the reviewer for pointing it out. We will fix colors and correct text in the revisions.

---

> > ### Author Rebuttal · Reviewer_o2yh · 2026-04-02
> >
> > I thank the authors for the rebuttal. Thank you for the clarification and for addressing the issue. This resolves my concern and I have no further questions. I also found the additional points raised by other reviewers helpful for further discussion. My overall assessment remains unchanged.

---

### Official Review · Reviewer_ReVR · 2026-03-12

**Soundness:** 4
**Presentation:** 3
**Significance:** 3
**Originality:** 2
**Overall Recommendation:** 4
**Confidence:** 4

**Summary:**

This paper builds on prior work studying OthelloGPT, which found that the model learns an implicit board-state representation when trained only on move sequences of Othello. It presents MetaOthello, a controlled suite of Othello variants with shared syntax but different rules or tokenisation. The authors use this suite to study how world models are organised in a shared representation space when a single model is trained on multiple variations of a the same game.

They train small GPT models on mixtures of Othello variants, each of which is defined by a starting board state, a validation rule, and an update rule. Specifically, they include NoMidFlip (updates only flip the outermost two flanked tiles), DelFlank (different initialisation, validation rule, and update rule). They also include Iago, which employs a bijection that maps an arbitrary token to a physical move. These maps differ between games.

They find that models do not partition their representational capacity into isolated subspaces, but converge on a mostly shared board state representation. This is validated by measuring the alignment of probe weights trained on different game variants, as well as performance of causal interventions of different probes. For Iago (where tokenisation is scrambled), they find that the model learns the same latent structure, differing only by a rotation induced by the input vocabularies. However, for ambiguous sequences (valid for multiple game variants) the representations diverge in proportion to where the game rules actually conflict.

**Compliance With Llm Reviewing Policy:**

Affirmed.

**Final Justification:**

The rebuttal, while adding interesting points about the mechanisms a model uses to determine the game variant that is being played and therefore strengthening the paper, has not significantly changed my assessment. The paper presents technically solid work but, I believe, still best fits the descriptions of "weak accept". It's potentially useful for others in the mechanistic interpretability field, but, as reviewer 6GbT pointed out, most work on studying world models in small, synthetic transformers was done in 2022 & 23 and will thus likely have low impact.

**Key Questions For Authors:**

1. The model trained here is relatively small. Is the shared representation space only an artefact of the small model size (limited representation capacity)? Would a sufficiently large model maintain fully separate world models?
2. Layer 5 appears to be involved in identifying the game variant that is played. But how does game identification work mechanistically?

**Limitations:**

Yes

**Strengths And Weaknesses:**

**Strenghts**
- The paper evaluates the alignment of the shared representations using multiple techniques: probe weight alignment, probe generalisation, and generalisation of causal interventions.
- The fact that the representations are aligned, up to a rotation, in a setting with similar rules but different tokenisation is interesting.
- The authors carefully discuss the implications (and limitations) of experimental results.
- MetaOthello, as a controlled setup for studying shared representation spaces, could be useful to future work.

**Weaknesses**
- The hypothesis that models would build isolated world models for conceptually very similar games was, arguably, the less likely alternative and is thus not surprising.
- One of the most interesting question remains unanswered: The representations appear causally interchangeable (i.e. are mostly shared between different game variants) for unambiguous sequences but also encode different information for ambiguous sequences where game rules conflict. The authors stop short of investigating how the model implements both simultaneously and instead brush this off in the conclusion by pointing to limitations of linear analysis.

---

> ### Author Rebuttal · Authors · 2026-03-31
>
> The reviewer identifies the core open questions in our work. We address each below:
>
> **W1.** There are a few points that we would like to clarify:
> 1.  Our setup allows us to define conceptual similarity by reference to the generative rules. Classic and Nomidflip are more conceptually similar (the only difference being that Nomidflip does not flip middle tiles). Classic and Delflank, however, are far more dissimilar (new validation and update mechanism), as evidenced by the low overlap in their game trees, and dissimilarities of their corresponding probe dimensions. In our view, the surprising result is not merely that similar games (Classic/NoMidFlip) share world models, but that very different games (Classic/DelFlank) also converge on a causally shared model.
> 2. More generally, the question of models converging on one representation for more-or-less different tasks is hard to pose precisely in the NLP settings, where we lack a ground truth representation and rule set to compare against. We argue that posing this question in an environment that allows for objective comparison to the ground truth itself constitutes a contribution.
>
> **Q1.** We have two reasons to believe that capacity is not a strong driver for the results:
> 1. The minimum representational budget for both games' board states is 2 × 192 = 384, leaving substantial headroom in the 512-dimensional space. Despite this capacity, the model still converges on shared representations.
> 2. Yuan and Søgaard (2025) show that similar world-model representations emerge across different model sizes for Othello, including models far larger than ours. Their results indicate that changing model size changes underlying representations more quantitatively than qualitatively.
>
> We can infer from this, to some extent, that changing model size should not impact the geometry of individual underlying representations. However, exploring this question of how model sizes impact shared representations is something we would like to investigate in future work.
>
> **W2, Q2:** These are important open questions, and we appreciate the reviewer's emphasis on them. In attempting to answer them for ourselves, we have found evidence that the two are linked: both the game identification, and game-specific deviations from the shared board representation, are computed via a specific routing circuit in layers 4 and 5. We summarize the new findings below and will integrate them into the revised manuscript:
>
> **Where the model calculates game identity.** We sample ambiguous prefixes (sequences valid under both Classic and NoMidFlip) and append a single disambiguating move. We then track the signed margin of the game-ID probe and decompose the results into attention and MLP contributions at each layer. The game-id margin is near-zero through layers 1-3 and rises at layers 4-5. At layer 4, the gain is dominated by the MLP (attn gain ≈ 0.03, MLP gain ≈ 1.32). At layer 5, attention contributes the most (attn gain ≈ 4.06), followed by MLP (MLP gain ≈ 1.21). Ablation confirms causal necessity: zeroing the layer-4 MLP output produces the largest drop in the layer-5 game margin (Δ = −3.80), while zeroing head 5.5 also produces a significant drop (Δ = −1.81). **The primary circuit is therefore: Layer 4 MLP → Head 5.5 → Layer 5 MLP.**
>
> **What the Layer 4 MLP computes.** We find that the Layer 4 MLP reads the shared board representation built by earlier layers and computes a rule-conflict feature: for each tile, it increases the margin for the correct game's tile state over the alternative game's tile state. On tiles where Classic and NoMidFlip produce different board states ("conflict tiles"), the mean margin gain is 2.31, compared to 0.0 gain for non-conflict tiles. Board-state heatmaps, which we will include in the revised manuscript, confirm the MLP signal is spatially concentrated on exactly the conflicting tiles. **The shared board representation is therefore compatible with the computational substrate that does game identification.**
>
> **Causal sufficiency.** We use the discovered circuit to steer ambiguous prefixes to play one game over the other. At scale 2.0, steering the full routing circuit (noted above) in the Classic direction shifts the share of Classic-unique tokens from 0.49 to 0.89, while NoMidFlip-unique tokens drop to 0.01. Shared-move mass stays roughly constant (~0.79) across all interventions, confirming the circuit specifically reallocates probability between game-unique branches. We also control for recency effects by comparing results when an additional ambiguous move (rather than a disambiguating move) is appended, confirming that the circuit responds to rule conflicts, not to recent tokens.
>
> These findings provide a substantial, if not complete, resolution to the motivating tension. The model linearly represents a single board state. It uses this information in conjunction with other mechanisms described above to calculate conflicting tiles between Classic and NoMidFlip.

---

> > ### Author Rebuttal · Reviewer_ReVR · 2026-04-03
> >
> > Thanks for the detailed response and follow-up experiments around the mechanisms a model uses to determine the game variant that is being played. I have no further questions.

---

### Official Review · Reviewer_Bc18 · 2026-03-12

**Soundness:** 4
**Presentation:** 3
**Significance:** 3
**Originality:** 3
**Overall Recommendation:** 4
**Confidence:** 3

**Summary:**

This paper studies language model representations, specifically how language models represent multiplied conflicting rule sets (i.e. world models). They choose a controlled setting focusing on different variants of othello to characterize the structures learned. Through this analysis they find a number of interesting insights about how language models represent information.

**Compliance With Llm Reviewing Policy:**

Affirmed.

**Final Justification:**

The authors have addressed W1-W4, their argument for W5 is reasonable but without empirical support doesn't warrant an increase in score. I'll maintain my already positive review of the submission.

**Key Questions For Authors:**

(Q1) Why in figure 4 does the layer 8 intervention appear to have very high alpha at move 0 before dropping for nearly all other moves, while all other layer interventions seem to have the opposite trend going from move 0 to move 1,2,3?

(Q2) What are the practical implications of these results? Is there evidence these findings reproduce for real language models?

**Limitations:**

Yes

**Strengths And Weaknesses:**

Strengths

(S1) Othello variants are an elegant choice for their experiments, enabling them to ablate the structure of the problem in very controlled ways (e.g. game rule changes, surface form changes).

(S2) Alpha metric is independent of the irreducible entropy of a sequence s and also penalizes random guessing.

(S3) Mixed game models share earlier layer representations and only diverge later in the network. Intuitive the network might learn this structure but nice to show it so concretely.

(S4) The syntax invariance experiment finding that a rotation will align board representations is a really interesting insight about how language models represent different tokenizations of the same game.

(S5) Study of steering identifies layer 5 as a phrase transition in the representation.

Weaknesses


(W1) Figure 6a is a bit hard to parse. For example “1” should be a black line? But I don’t see a black line.

(W2) The notation P is overloaded, one page 2 it is used to indicate the power set of M, but on page 3 it is the ground truth distribution.

(W3) The writing in the experiments section is quite dense in places and could be more explanatory. For example, I wasn’t sure why the probe output dimension is 192?

(W4) This work depends heavily on Procrustes alignment for some experiments, some discussion of what it is and how is works would be helpful.

(W5) Some application of this analysis to real language models would significantly strength this work.

---

> ### Author Rebuttal · Authors · 2026-03-31
>
> We thank the reviewer for their thoughtful feedback and careful reading of our work. We address their concerns point by point:
>
> **W1:** Thank you for pointing out this mistake. There is a cosmetic error in Figure 6a. We reduced the opacity for layers other than 2 and 3 (to highlight the main result), but failed to do so for the layer indicator markers. We will correct this error in the manuscript.
>
> **W2:** Thank you for catching this. We will change the power set notation on page 2 to $\mathcal{P(M)}$ to distinguish it from the probability distribution P used elsewhere, and add a note at first use to prevent confusion.
>
> **W3:** The probe output dimension is 192 because it predicts 64 board tiles x 3 states (mine, yours, empty). We acknowledge that the writing in the methods is a bit dense, in the revision we will clarify the experimental setup and work to make the text clearer.
>
> **W4:** This warrants more explanation, and we will add a dedicated description in the revision. Orthogonal Procrustes finds the orthogonal matrix R that best aligns two matrices A and B by minimizing $\|AR - B\|_F$
> which is solved in closed form via the singular value decomposition (SVD) of $B^T$. Intuitively, it asks: can the two sets of learned directions be made to coincide by a rigid rotation or reflection of the representation space, without stretching or distortion? High post-alignment similarity indicates that the two representations encode the same features in the same geometry, differing only by an arbitrary choice of coordinate axes. We will include this explanation alongside Figure 2 in the revision.
>
> **W5, Q2:** These are important questions. We cannot directly test our findings on large language models here; MetaOthello is designed as a controlled precursor precisely because "ground-truth" latent states are inaccessible in real large-scale LMs. However, we expect these findings to scale for several reasons:
> 1. **Platonic Convergence.** Our results align with the Platonic Representation Hypothesis, which posits that diverse training naturally drives networks toward isomorphic representations. (Huh et al., 2024)
> 2. **LLM Equivalents.** Emergent, linear state-tracking (world models) have already been observed in LLMs for concepts like space and time (Gurnee & Tegmark, 2023). And world-model representations have been shown to generalize across architectures and scales for both Othello (Yuan & Søgaard, 2025) and chess (Karvonen, 2024).
> 3. **Task Composition.** Larger models organize distinct skills via task vectors (Ilharco et al., 2023) and function vectors (Todd et al., 2024). MetaOthello grounds this by showing exactly how conflicting multi-task representations are dynamically arbitrated at the layer level.
>
> In large scale language models, it is important to know how the model represents, composes, and balances competing and diverse tasks. We see MetaOthello as a controllable complement to research such as the functional task vector paper cited above. Using this framework, we can make precise the notions of “task similarity” and context-dependent task ambiguity. We can then develop testable hypotheses for language models guided by mechanistic principles grounded in this toy model. For example, we find that the directions controlling task identity emerge in later layers when the tasks are more similar. We can now validate this, in future work, on real examples like coding in different languages and different tasks in the same language. Our findings should then inform steerability of these models for these tasks.
>
> **Q1:** First, a minor clarification that we will also correct in our figure: the move 0 corresponds to next token prediction accuracy given only one move. Moving on, we address the core of the question-
>
> This pattern stems from how differently layers handle the Procrustes rotation (Ω) with and without cross-attention:
> - Move 0 (Single-token, no cross-attention): Blocks 1–7 are highly equivariant to Ω. Block 8 is not, because its token-to-logit mapping relies heavily on exact surface tokens when prior context is absent.
> - Moves 1+ (Cross-attention active): Early-layer equivariance drops (as Q⋅K interactions do not perfectly commute with Ω), while block 8 becomes highly robust due to the newly available, richer game-state context.
>
> This asymmetry explains the α dynamics. At move 0, interventions at layers 1–7 must pass through the non-equivariant block 8, which suppresses their α. A layer-8 intervention bypasses this bottleneck entirely, hitting the unembedding directly for a high initial α. Once cross-attention activates at move 1+, block 8 becomes robust, allowing the early-layer interventions to recover. Layer 8's own α gradually declines thereafter because Ω does not perfectly commute with the final linear unembedding where the mismatch compounds as the model's next-move distributions become more peaked later in the game.

---

> > ### Author Rebuttal · Reviewer_Bc18 · 2026-04-03
> >
> > The authors have addressed W1-W4, their argument for W5 is reasonable but without empirical support doesn't warrant an increase in score. I'll maintain my already positive review of the submission.

---

### Official Review · Reviewer_6GbT · 2026-03-24

**Soundness:** 3
**Presentation:** 4
**Significance:** 2
**Originality:** 2
**Overall Recommendation:** 3
**Confidence:** 3

**Summary:**

This paper introduces a new dataset called MetaOthello (consisting of several variants of Othello) in order to study how transformer models learn game rules across different variants. The authors frames this as allowing them to study how multiple world models can be learned in transformers. After training GPT-style models on a mixture of these different variants, they use linear probes to recover the internal board representation from these transformers and then perform a bunch of analyses. The key findings include that transformers trained on mixed game data converge on shared board state representations and that representations for isomorphic games are equivalent up to orthogonal rotations.

**Compliance With Llm Reviewing Policy:**

Affirmed.

**Final Justification:**

I think this paper has a lot of good ideas and the analyses performed are interesting. However, I still have concerns are around 1) the fairly loose use of 'world model' in the paper (despite the experiments being limited to just learning the deterministic rules of Othello), and 2) the limited domain of Othello to test these methods on. I would have liked to see more domains/variants tested so that the findings from this paper can be useful to the broader community working on world models. Overall, I'll maintain my rating of weak reject.

**Key Questions For Authors:**

I would love to see the authors' responses to the weaknesses I listed above.
My main questions for the authors revolve around the following:
1. I feel Othello is a limited environment to test the hypotheses that the authors want to test in the first place. I feel like you could use more games, not just Othello.
2. I also think using the term "world model" requires some serious rethinking because it means something specific in the RL literature (plus the paper only considers pairs of game variants). I am not completely convinced that just recovering the board state after a sequence of moves completely encapsulates the notion of a world model in general.

**Limitations:**

yes

**Strengths And Weaknesses:**

Strengths:
* The experiments performed in the paper are good and the paper is presented well.
* The idea of creating multiple variants of the game to test internal representations is novel

Weaknesses:
* I'm not an expert in mechanistic interpretability but the main weakness of this paper in my opinion is the limited domain on which the analyses are performed. I like the idea of having multiple Othello worlds but just having four variants seems small to me and moreover Othello is also a fairly limited game. So even though the results are interesting, I'm not sure if they are strong enough for me to assume they would hold in a new domain. I would have liked to see the same idea, i.e having variants of the same game be applied to perhaps more games (e.g chess, backgammon, etc.).
* Related to the above I wonder if the authors could test out more variants of Othello? I like the fact that mixing the different variants allows us to figure out how the shared representations are learned so doing this on more than four would be interesting
* The result around transformers learning a shared representation across the different variants didn't seem very surprising to me. Isn't this basically what we have seen in other domains? For example in NLP, language modeling allows transformers to learn a shared representation of language that can then be useful for many tasks that use language and other tasks that are even not natural language (e.g. coding). Similar for policy learning across diff environments in RL. I know this paper is about world modeling but I don't see why that cannot be inferred from what we've seen transformers do in other domains.
* Finally, I personally don't think learning the rules of Othello is strong enough to be called a "world model" in general. I would love to see some more stochasticity in the transitions of the environment chosen; that makes it more interesting to perform a study with this kind of game variants. For example, Nethack it's a good game with decent stochasticity- one could could consider varying the rules of various NetHack to create different variants.

---

> ### Author Rebuttal · Authors · 2026-03-31
>
> This is substantive feedback and we address each concern enumerated below:
>
> **W1, W4, Q1, Q2:** These points are interrelated and together point to a single clarification we should make more prominent in the paper.
>
> **Why Othello.** Following work by Li, et al. (2022) and Nanda, et al. (2023), *Othello has become a canonical testbed for “world model” interpretability.* MetaOthello builds on this foundation: because we already understand Othello-GPT in detail, we can isolate what changes when a second rule system is introduced.
>
> **Why not Chess or Nethack.** Our contribution depends on our ability to tune the similarity between game variants using the same board geometry: Classic and NoMidFlip diverge slowly whereas Delflank diverges early. Chess has a vastly larger game tree and a more complex base game, which is harder to  perturb in a controlled way (though mechint work on chess does exist; Karvonen, 2024). Using variants of a game like NetHack risks conflating stochasticity with uncertainty over the (deterministic) generating rules, making it harder to make precise mechanistic and causual claims.
>
> **On "world model" terminology.** We recognize that “world model” has many different definitions across machine learning, neuroscience, cognition, etc. As the reviewer points out, the specific meaning in the RL literature, as illustrated by Ha & Schmidhuber, 2018; Hafner et al., 2023, typically implies a learned dynamics model used for planning. We follow the mechanistic interpretability definition established by Li et al. (2025, "What does it mean for a neural network to learn a 'world model'?"): a causally complete internal representation sufficient to reconstruct the latent state of the data-generating process. The models we study meet this standard; the question is how they organize world models of conflicting generative processes. We will expand the current footnote into a dedicated subsection in the revision to make this distinction explicit.
>
> **W2:** We appreciate the encouragement and agree with the reviewer that experiments with more variants could be enlightening, although not possible within the rebuttal window. Moreover, we believe the core tension present in these variants—how the model simultaneously represents a single board state while accurately predicting the union of two games—itself remains to be resolved. We present our attempts at that resolution in our response to reviewer ReVR. We certainly plan to test out more variants in future work, particularly as we refine our hypotheses about the dominant modes of variance between the learned representations of different games. For instance, one question we may want to ask is: can we design variants of Othello that break the causal equivalence of their board representations? What distinguishes these from variants such as DelFlank and NoMidFlip?
>
> **W3:** This is a helpful clarification. The idea that transformers build shared representations (even a linear one) is not in itself surprising (for example, as shown in Vafidis et al., 2025). What’s novel about MetaOthello is not the shared representation itself, but that the structure of the shared representation can be compared against our exact knowledge of the underlying generative processes. In natural language, we do not know what the underlying process is to be represented, so we cannot compare the learned representation to a “ground truth.” Our core contribution is to take a known ground truth (Othello) that has been used in world model studies before, and alter it to include a well-defined notion of diversity in the generative mechanisms. This leads us to results that could not have been found without a diverse mixture of ground truth processes, such as:
> - The model accurately, linearly, and causally represents the posterior probability of each generating process given the inputs.
> - The state representations for two very different processes are causally equivalent, even though the model accurately distinguishes the two dynamics in its outputs.
> The revision will reorder the contributions to lead with the mechanistic results rather than the shared-representation finding to clarify the primary contribution.
>
> **Conclusion:**
> We thank the reviewer for their deep engagement with our work and for prompting these important clarifications regarding the novelty of our claims and our use of terminology. We plan to update the manuscript with the following revisions to ensure these points are front and center:
> 1. Add more information about our use of the world model terminology and why it is relevant to our work and broader literature.
> 2. More explicitly outline our contributions as a methodological development for mechanistic interpretability and downstream experimental impacts for multi-task representations.
>
> We believe these revisions will significantly clarify the scope and value of our contributions, and we hope this response addresses the reviewer's core concerns.

---

> > ### Author Rebuttal · Reviewer_6GbT · 2026-04-01
> >
> > Thanks to the authors for providing a detailed response to my questions - you're right in that my core concerns are around 1) the fairly loose use of 'world model' in the paper, and 2) the limited domain of Othello to test these methods on. Elaborating more below.
> >
> > 1. First, thanks for providing the reference to the Li et al. (2025, "What does it mean for a neural network to learn a 'world model'?") work defining a world model in this context. I took a quick look at it and it is an unpublished paper on arxiv (withdrawn from a previous conf). While that in itself is not a deal breaker, I actually looked at the reviews for the paper and several points there match my thoughts exactly. The definition itself seems very vague, so if you are basing your work on that particular unpublished definition, I don't think that's necessarily a solid basis to use.
> > That said I don't necessarily mean that what you are trying to do is not a world model. It is definitely a world model but a very simplistic one simply because there is no stochasticity involved, so it feels very "toy". To be explicit my initial comment "don't think learning the rules of Othello is strong enough to be called a "world model" in general." is meant to say that simply testing these methods on Othello and then having world model in the title of your paper seem at odds with each other. Ideally you have a few other environments where there are a variety of world models and you show similar results or a combination of results.
> >
> > (more of a meta-comment) Finally, the papers you're referring to in the previous work on Othello are from 2022 and 2023. We are currently in 2026 so I think the bar is higher now and scale of things you could accomplish is definitely way more than what was done in the previous paper (esp. with AI coding tools, etc.). I think you can definitely aim for bigger things! (mean this purely from a constructive manner, don't take it any other way). I personally, for one, would find this paper's experiments done on a few other environments to be an excellent overall story (regardless of outcome).
> >
> > 2. Thanks for the response to W3. That definitely helps a bit although I think the works in RL also have access to the ground truth trajectory so they should be able to do a similar kind of comparison (e.g. https://arxiv.org/html/2506.13958v1). So I'm not sure I fully buy this argument.
> >
> > Overall I just want to say I think there's a lot of excellent technical work in this paper. The amount of analysis carried out within the MetaOthello domain is significant and definitely quite interesting (so great work on that!). I just felt a bit disappointed that it's on a very limited domain and my expectation when looking at the title was not what I got from the paper, hence my comments above. I agree with your comments that some of these changes may not be possible within the rebuttal window, and may even make the paper a completely different version to what is currently submitted.

---

> > > ### Author Response · Authors · 2026-04-07
> > >
> > > We thank the reviewer for their generous characterization of our technical work and their continued engagement with the underlying philosophical tensions of the paper. "World model" is used broadly across the field and carries a heavily debated formal meaning in machine learning. This is precisely why these discussions are exciting and productive for the community. We now address the more philosophical questions the reviewer raises.
> > >
> > > **World Model definition.** Li et al. (2025) is indeed unpublished, but our use of the term does not rest on that paper's authority alone. Our usage follows an established tradition in the field [1, 2, 3, 6]. But we think the reviewer is gesturing at something deeper. The question is whether the kind of internal state-tracking we study in Othello deserves to be called "world modeling" at all, or whether the term should be reserved for richer, more stochastic, more ecologically diverse settings.
> > >
> > > As Mitchell (2025) [4] observes in her survey of the debate, "world model" does not have a single, agreed-upon definition in AI. She identifies a spectrum ranging from static lookup tables to fully simulatable causal models, and notes the community is split on whether sequence models can learn world models at all. Several recent formalizations converge on our effective use of the term. Millière & Buckner (2024) [5] define world models as "structure-preserving [...] causally efficacious representations" of an input domain. (See also Spies et al., 2024 [6]) These definitions share a common core with the Othello-GPT literature: an internal representation that is sufficient to reconstruct the latent state, and causally implicated in the model behavior proven through intervention. As we discuss next, the meaning of "world model" in a reinforcement learning context differs in a subtle but, we think, important way. A world model in this context means something more akin to a coarse-grained representation of the world sufficient for the agent to use for policy optimization [8, 9, 10]; this will sometimes but not always overlap with a representation that is sufficient to reconstruct the causally relevant latent state [7].
> > >
> > > In brief, RL questions if we can build a model that learns useful dynamics; mechinterp questions whether a model learned a representation that reflects the causal structure of the generating process. In MetaOthello, we go a step further. We ask: when the generating process is a mixture of causal structures, how does the model organize its internal representations to track them simultaneously?
> > >
> > > **Usefulness.** We appreciate the reviewer's pointer to the offline RL literature [11], but we think the cited comparison actually highlights the value of toy models. The referenced paper analyzes learned embeddings through aggregate geometric statistics, covariance structure, vector magnitude, and cosine similarity, and relates those statistics to task performance; it does not probe for specific representations of a latent environment or test its causal efficacy by intervention. This is a natural feature of RL settings: what the model needs to represent is determined by the reward function rather than the full causal structure of the environment. Toy domains like Othello remove that confound, letting us ask precisely what latent state is represented, how, and how it changes when multiple generative processes share a state space but differ in transition rules.
> > >
> > > This has been a genuinely productive exchange and we thank the reviewer for pushing us to sharpen these positions. We will incorporate points from these discussions in our revised draft to better contextualize the scope and history of our project.
> > >
> > > **References**
> > > - [1] Gurnee, W. & Tegmark, M. (2023). Language Models Represent Space and Time. arXiv:2310.02207.
> > > - [2] Karvonen, A. (2024). Emergent World Models and Latent Variable Estimation in Chess-Playing Language Models. arXiv:2403.15498.
> > > - [3] Yuan, Y. & Søgaard, A. (2025). Revisiting the Othello World Model Hypothesis. arXiv:2503.04421.
> > > - [4] Mitchell, M. (2025). LLMs and World Models, Parts 1 & 2. AI Guide Substack.
> > > - [5] Millière, R. & Buckner, C. (2024). A Philosophical Introduction to Language Models. arXiv:2401.03910.
> > > - [6] Spies, A.F., et al. (2024). Transformers Use Causal World Models in Maze-Solving Tasks. arXiv:2412.11867.
> > > - [7] Richens, J., et al. (2024). Robust Agents Learn Causal World Models. ICLR 2024. arXiv:2506.01622.
> > > - [8] Ha, D. & Schmidhuber, J. (2018). World Models. arXiv:1803.10122.
> > > - [9] Hafner, D., Lillicrap, T., Ba, J., & Norouzi, M. (2020). Dream to Control: Learning Behaviors by Latent Imagination. ICLR 2020. arXiv:1912.01603.
> > > - [10] Moerland, T. M., Broekens, J., Plaat, A., & Jonker, C. M. (2020). Model-based Reinforcement Learning: A Survey. arXiv:2006.16712.
> > > - [11] Guiducci, L., Rizzo, A., & Dimitri, G. M. (2025). Toward Explainable Offline RL: Analyzing Representations in Intrinsically Motivated Decision Transformers. arXiv:2506.13958.

---

### Decision · Program_Chairs · 2026-04-30

**Decision:**

Accept (regular)

**Comment:**

This paper introduces MetaOthello, a collection of Othello game variants that use the same syntax but have different rules or tokenizations. The paper trains small GPT models on these game variants to analyze how the model organizes multiple world models in a shared representation space. The authors find that, rather than learning isolated sub-models, transformers trained on the MetaOthello mixed game data learn a shared board-state representation that transfers across game variants.

Reviewer 6GbT found the paper to be well-written. However, this reviewer raised several concerns regarding the limited scope of the paper, only studying specific variants of Othello. The reviewer noted that the paper studies only four variants of the game, and mentioned that Othello is a fairly limited game. The reviewer suggested expanding the scope to include other types of games at least.

The reviewer also noted that learning the deterministic rules of Othello may not be sufficient to constitute the “world model” terminology used in the paper, and suggested expanding the analysis to games with more stochasticity in the transitions, such as Nethack.

The authors’ rebuttal points out that the original Othello game is a testbed for world model interpretability, and MetaOthello builds on this with a set of Othello variants.

Overall, most of the reviewer’s concerns were addressed by the rebuttal, and the remaining core concerns are mostly regarding whether the scope of the work is sufficient (only focusing on one game, Othello). I think that taking everything into consideration, this paper makes a sufficient contribution.

Reviewer Bc18 found that Othello was a good choice of game for the empirical investigations, as it enables various types of ablations with respect to the game rules and surface forms. This reviewer also appreciated the detailed investigation into early vs late layer representations. The main concerns raised by this reviewer are regarding the clarity of the exposition and the lack of application of this analysis to real LLMs. The authors’ rebuttal addressed most of the concerns raised by this reviewer. However, the concern about the implications for real LLMs remained after the rebuttal.

Reviewer ReVR found the findings of the paper regarding the fact that representations are aligned up to a rotation to be interesting, and the discussion to be comprehensive. This reviewer also found that MetaOthello could be useful for future work, as it provides a controlled setup for analyzing shared representation spaces.

Reviewer ReVR raised a concern regarding unanswered questions about representations. The reviewer also raised a concern regarding the small size of the model used, and wondered whether a larger model would behave differently (e.g., maintaining separate world models). The rebuttal addressed most of the reviewer’s concerns.

Overall, this reviewer believes that the paper will have moderate to low impact and will mostly be interesting for people studying mechanistic interpretability. The reviewer kept their score of 4.

Reviewer o2yh found that the paper was well-written, rigorous in its empirical investigation, and significant (while acknowledging that it only studies specific Othello game variants). This reviewer did not identify any major weaknesses.

In sum, while the scope of the paper is limited, the analysis seems to be sufficient to be of interest to the mechanistic interpretability community, and thus I recommend acceptance.